# Cooperative Spectrum Sensing Based on Multi-Features Combination Network in Cognitive Radio Network

**DOI:** 10.3390/e24010129

**Published:** 2022-01-15

**Authors:** Mingdong Xu, Zhendong Yin, Yanlong Zhao, Zhilu Wu

**Affiliations:** School of Electronics and Information Engineering, Harbin Institute of Technology, Harbin 150001, China; 18b905032@stu.hit.edu.cn (M.X.); 16B905027@stu.hit.edu.cn (Y.Z.); wuzhilu@hit.edu.cn (Z.W.)

**Keywords:** cooperative spectrum sensing, cognitive radio network, deep learning, large dynamic signal-to-noise ratio

## Abstract

Cognitive radio, as a key technology to improve the utilization of radio spectrum, acquired much attention. Moreover, spectrum sensing has an irreplaceable position in the field of cognitive radio and was widely studied. The convolutional neural networks (CNNs) and the gate recurrent unit (GRU) are complementary in their modelling capabilities. In this paper, we introduce a CNN-GRU network to obtain the local information for single-node spectrum sensing, in which CNN is used to extract spatial feature and GRU is used to extract the temporal feature. Then, the combination network receives the features extracted by the CNN-GRU network to achieve multifeatures combination and obtains the final cooperation result. The cooperative spectrum sensing scheme based on Multifeatures Combination Network enhances the sensing reliability by fusing the local information from different sensing nodes. To accommodate the detection of multiple types of signals, we generated 8 kinds of modulation types to train the model. Theoretical analysis and simulation results show that the cooperative spectrum sensing algorithm proposed in this paper improved detection performance with no prior knowledge about the information of primary user or channel state. Our proposed method achieved competitive performance under the condition of large dynamic signal-to-noise ratio.

## 1. Introduction

In recent years, wireless communication technologies developed rapidly, including but not limited to the Internet of Things (IOT), 5G Networks, Internet of Vehicles (IOV) [1,2,3,4,5,6], and so on. With the rapid growth of wireless communication data traffic, especially the large-scale applications of IOT, IOV and the rapid development of 5G Networks, the demand for spectrum resources in wireless communication systems is increasing. The demand for spectrum resources of mobile communications systems would reach 1280–1720 MHz by 2020, which is predicted by International Telecommunication Union (ITU) [7]. According to statistics from the Federal Communications Commission (FCC), the existing spectrum allocation methods make the utilization of spectrum resources insufficient [8]. Cognitive radio (CR) [9,10,11] is proposed as a key technology to improve the utilization of radio spectrum, which is essential to alleviate the scarcity of spectrum resources. In cognitive radio network (CRN), the temporarily unoccupied spectrum for the primary user (PU) can be accessed by secondary user (SU) uses opportunistic method. Therefore, the primary task of CR is to determine the existence of PU, and spectrum sensing technology is the basis of CR. When the CRN is in a relatively harsh communication environment, the signal-to-noise ratio (SNR) of the signal received by the SU from the PU is very low. At this time, the SU still needs to accurately detect the PU and avoid using its channel, otherwise it will cause huge interference to the PU. Hence, it is urgent to enhance the generalization performance of spectrum sensing algorithm under large dynamic SNR. More energy consumption will be generated as the secondary users of the 5G Networks have to search broader bandwidths to gather spectrum information. As a result, efficient spectrum sensing technology is urgently needed at present.

To improve the utilization of radio spectrum, many research works were performed. Conventional spectrum sensing methods for single-node spectrum sensing include matched filter detection [12,13], cyclostationary detection [14,15], and energy detection [16,17], etc. Matched filter detection requires the secondary user receiver to fully understand the primary user signal (e.g., bandwidth, operating frequency, modulation type, etc.). Cyclostationary detection requires the secondary user receiver to know the periodicity of the primary user signal. Energy detection can be achieved with no prior knowledge, but it is susceptible to noise. However, practice proved that traditional single-node spectrum sensing techniques are susceptible to environmental shielding and transmission fading.

Cooperative spectrum sensing (CSS), as a key technology to improve the reliability of spectrum sensing, was widely studied [18,19,20]. Compared with that of single-node spectrum sensing, CSS merges the data of different sensing nodes through the fusion center to improve detection reliability. According to the different ways in which SUs participate in collaboration and exchange information with each other, CSS is divided into three categories, namely: centralized cooperation [21], distributed cooperation [22], and relay cooperation [23]. Due to the characteristics of low algorithm complexity and less computationally intensive, centralized cooperative spectrum sensing was applied extensively. The data fusion algorithm of conventional CSS scheme, such as energy detection is divided into two types, namely: hard decision (HD) [24] and soft decision (SD) [25]. Moreover, in the HD scheme, common decision-making rules include OR rules, AND rules, and K-OUT-N rules. Compared with the way that the perception result sent by SU be a binary decision result (HD) of 0 or 1, the better performance by sending credibility soft information (SD) [26]. However, the consequent impact is that HD has more communication overhead. Hence, choosing the suitable data fusion method is a compromise between perceived accuracy and communication overhead. With the mass development of deep learning algorithms, many research works based on deep learning algorithms for CSS were proposed [27,28]. Ref. [27] proposed deep cooperative sensing, which uses convolutional neural networks (CNNs) to achieve cooperative spectrum sensing and achieved better results than other algorithms. Moreover, ref. [28] proposed a joint CNN and Long Short-Term Memory (LSTM) detection algorithm to exploit the soft information from distributed sensing nodes. In addition, the CSS schemes based on DAG-SVM algorithm were proposed in [29], which achieved good results compared with that of other machine-learning-enabled solutions.

In this paper, we propose novel CSS schemes based on CNN and gate recurrent unit (GRU), in which CNN is used to extract spatial features and GRU is used to extract the temporal feature. The novelties and contributions of this paper are summarized as follows: Since the task characteristics of cooperative spectrum sensing, we proposed a CSS scheme employing CNN and GRU, which can effectively improve detection performance in low SNR. Aiming at the hard decision of traditional centralized cooperative spectrum sensing schemes that cannot exploit confidence information, we propose “Combination-net” to utilize the information of each node. We conducted 8 kinds of modulation signal experiments for the proposed model to adapt to detecting various radio signals. The simulations show that effectiveness of the proposed model for multiple types of signals under harsh sensing conditions.

The remainder of this manuscript is organized as follows. Section 2 gives a detailed description of the system model for CSS in this paper. Section 3 describes the CSS scheme based on Multifeatures Combination Network (MCN) in this work. In Section 4, the performances of the proposed algorithm are shown with the simulation results and analysis. Finally, the conclusion of this paper is summarized in Section 5.

## 2. System Model

In this section, we introduce the system model of the proposed cooperative spectrum sensing problem. The key with spectrum sensing is to obtain the usage characteristics of a specific spectrum. We consider spectrum sensing is to perceive whether PU presences in a specific channel through SU. Spectrum sensing can be modeled as classic binary hypothesis testing problem, which is corresponding to the absence and presence of the PU respectively.
(1)$$\begin{array}{l} H_{0}:y(n)=v(n),\\ H_{1}:y(n)=x(n)+v(n), \end{array}$$
where y(n) is the received signal of SU, x(n) denotes the signal from PU, v(n) means the additive white Gaussian noise (AWGN).

To determine whether the PU is present, we compare the test statistic T(y) with the preset threshold λ, which can be expressed as:(2)$$\begin{array}{l} T\left(y\right)<\lambda \;\;\;H_{0},\\ T\left(y\right)>\lambda \;\;\;H_{1}, \end{array}$$
if the test statistic is less than the threshold, the PU is absent, as $H_{0}$, and if the test statistic is greater than the threshold, the PU is present as $H_{1}$, which can be expressed as:(3)$$\begin{array}{l} P_{d}=\mathrm{Pr}\left(T\left(y\right)>\lambda |H_{1}\right),\\ P_{f}=\mathrm{Pr}\left(T\left(y\right)>\lambda |H_{0}\right), \end{array}$$
where Pr(·) means the conditional probability, $P_{d}$ denotes the probability of detection, $P_{f}$ denotes the probability of false alarm for a single node.

The system model considered in this paper is shown in Figure 1, which comprises a primary user transmitter and a primary user receiver, several CR users or SUs, and a fusion center. In this work, the fusion center selects a channel and controls CR users to perform their own local spectrum sensing. Second, CR users upload the obtained sensing data through the reporting channel. Finally, the fusion center gathers the received sensing information to determine whether the PU exists and distributes the sensing results to the CR users. In CSS, data transmission and information fusion increase the energy consumption and detection delay of system. To enhance the generalization performance of spectrum sensing method under large dynamic SNR, optimizing fusion rules is an urgent issue in CSS.

The K-out-of-N criterion is a common form of hard-decision fusion criterion rule in CSS. If K or more users among the N SUs determine that the PU exists, the final decision is that the primary user exists; otherwise, it is deemed that the PU does not exist. The probability of detection $P_{d}^{\mathrm{css}}$ and probability of false alarm $P_{f}^{\mathrm{css}}$ can be expressed as:(4)$$\begin{array}{l} P_{d}^{\mathrm{css}}=\sum_{j=K}^{N}\left(\begin{array}{l} N\\ j \end{array}\right)P_{d}{}^{j}{\left(1-P_{d}\right)}^{N-j},\\ P_{f}^{\mathrm{css}}=\sum_{j=K}^{N}\left(\begin{array}{l} N\\ j \end{array}\right)P_{f}{}^{j}{\left(1-P_{f}\right)}^{N-j}, \end{array}$$
where N denotes the total number of SUs, j denotes the current number of SUs, and K denotes the number of SUs that determine the existence of PU which can be defined as:(5)$$K=\left\lceil \frac{N+1}{2}\right\rceil$$

## 3. Proposed Method

CSS was proven to be an effective way to improve detection performance. Comparing with single-nodes sensing, cooperative spectrum sensing exploiting distributed nodes to improve reliability in a cooperative manner. Compared with that of single-node spectrum sensing, the fusion center to fuse the sensing information of each node to obtain the final decision. However, the fusion center makes the final result based on the binary decision result, that the confidence information cannot be used. Motivated by this, we proposed a detection algorithm for CSS which exploits soft information in this paper.

### 3.1. Multifeatures Combination Network

CNN and LSTM have unique ways to extract features, and both show good performance in the corresponding tasks. CNN is used to extract local correlation features, and the LSTM is suitable for extract the temporal feature [30]. Refs. [31,32] connected CNN and GRU in series and achieved effective results. However, due to the different types of features extracted by CNN and GRU, the serial structure will result in the loss of information inevitably. Thus, the parallel structure of CNN and GRU with a proper feature combination method can extract different types of features of the original data. In [33], we proposed the parallel CNN-LSTM network for single-node spectrum sensing, and it was proven to be feasible. Since the structure of the GRU model is simpler than LSTM, it is more suitable for building larger networks to fulfil the mission requirements of CSS. Hence, the detection scheme for CSS based on CNN-GRU network is termed as “Multifeatures Combination Network”. Multifeatures Combination Network consists of “CNN-GRU” module and “Combination-net” module. The CNN-GRU network extracts the spatial and temporal features, and the combination network receives the features to achieve a multifeatures combination. The architecture of the Multifeatures Combination Network is illustrated in Figure 2.

In conventional sensing systems, the hard decision information from distributed nodes is combined using a specific fusion rule; “Combination-net” is designed as the fusion center for fusing information. The Combination-net consists of one concatenated layer and two FC layers. Through extensive cross-validation, the concatenated layer had 32 neurons, and two FC layers had 8 and 2 neurons, respectively. For each sensing node (CR1 … CRn), the CNN-GRU network is employed to obtain the sensing information from the primary signal. Afterwards, it is fed into the Combination-net, which directly learns the best fusion rule through training.

The design of the classifier has an indispensable position that affects the performance of the detection model. The softmax function is designed for classification problems, especially for two categories classification problems. It is the most popular output function for classification tasks. Cross-entropy loss is the loss function corresponding to the softmax function, which can be written as:(6)$$L=-\sum_{k=1}^{M}x_{k}\log\left(P_{k}\right),$$
where $x_{k}$ denotes indicative variable (0 or 1), $P_{k}$ denotes the predicted probability, M denotes the number of categories.

The setting of hyperparameters affects the performance of the algorithm, which is an integral part of network. We first chose activation functions ReLU according to the task type and determined the types of loss functions. Then, the hyperparameters are chosen empirically through extensive cross-validation and testing. The hyperparameters of MCN for CSS in this work are shown in Table 1.

### 3.2. Gate Recurrent Unit Network

The CNN is powerless in solving the input sequences that exist in the relevant context. The Recurrent Neural Network (RNN) is proposed to solve this problem, which feeds back the previous information to the current task. Due to RNN using the traditional gradient descent method for parameter learning, it will cause the problem of vanishing gradient.

LSTM and GRU are specifically designed to avoid such drawbacks by introducing the gate structures. Compared with LSTM, which has three gate structures: forget gate, input gate, and output gate, GRU has two gate structures: update gate and reset gate. From a computing point of view, GRU is more efficient and scalable. Moreover, from the perspective of the model, LSTM with more complex parameters has a higher risk of overfitting. The advantage of GRU is the simplicity of its model, and it is more suitable for building larger networks. The reason of we introduced GRU is that the task characteristics of CSS determine that its network model is more complex than single-node sensing. The architecture diagram of GRU is described as Figure 3.

The reset gate determines how much the new input information is combined with the previous memory, which is defined as:(7)$$r_{t}^{j}=\sigma {\left(W_{r}x_{t}+U_{r}h_{t-1}\right)}^{j},$$
where $r_{t}^{j}$ is the activation vectors of the reset gate, $W_{r}$
is the weight matrix, $x_{t}$ denotes the current input at the time t, j is the *j*-th unit, $U_{r}$ is the and recurrent weights, $h_{t-1}$ is the corresponding unit output, and σ denotes the activate function which can be defined as:(8)$$\sigma \left(x\right)=\frac{1}{1+e^{-x}}.$$

The update gate $z_{t}^{j}$ defines the amount of memory saved to the current time, which can be written as:(9)$$z_{t}^{j}=\sigma {\left(W_{z}x_{t}+U_{z}h_{t-1}\right)}^{j}.$$

${\tilde{h}}_{t}^{j}$ represents the new state, obtained by weighting the previous state, which can be written as:(10)$${\tilde{h}}_{t}^{j}=\tanh{\left(Wx_{t}+U\left(r_{t}⊙h_{t-1}\right)\right)}^{j},$$
where ⊙ is an element-wise multiplication.

tanh(·) function is defined as:(11)$$\tanh\left(x\right)=\frac{e^{x}-e^{-x}}{e^{x}+e^{-x}}.$$

$h_{t}^{j}$ will retain the information of the current unit and pass it to the next unit, which can be written as:(12)$$h_{t}^{j}=\left(1-z_{t}^{j}\right)h_{t-1}^{j}+z_{t}^{j}{\tilde{h}}_{t}^{j}.$$

### 3.3. One-Dimensional Convolutional Neural Network

Compared with traditional algorithms, the neural networks are able to extract features and maps to values. However, conventional neural networks have the problem of excessive number of parameters. CNN solves the issue by adding a feature learning part. CNNs made unprecedented achievements in the field of Semantic segmentation and image processing. Although CNN is generally used for two-dimensional (2D) data processing, one-dimensional (1D) data processing is possible. According to different task requirements, 1D-CNN is applied in the field of speech recognition and natural language processing.

The schematic diagram of 1D-CNN is described as Figure 4, it consists of input layer, convolutional layer, pooling layer, fully connected layer, and output layer.

The Convolutional layer has two characteristics: sparse connection and weight sharing. Sparse connection solves the problem of excessive number of parameters in fully connected neural networks, and weight sharing effectively reduces network overfitting. The role of convolution layer is to perform convolution calculations and extract data features. The pooling layer carries out down-sampling on convolution feature map to reduce feature dimension and improve robustness. The purpose of the fully connected layer is to complete tasks through the extracted features.

## 4. Experiments and Results

### 4.1. Dataset Generation

In this paper, the dataset consists of Amplitude Shift Keying (ASK), Phase Shift Keying (PSK), and Quadrature Amplitude Modulation (QAM) and Linear Frequency Modulation (LFM). 

The mathematical expression of ASK signal can be expressed as:(13)$$x(t)=[A\sum_{n}a_{n}g(t-nT_{s})]\cos(2\pi f_{c}t+\theta ),$$
where A means the carrier amplitude, $a_{n}$ denotes the digital baseband signal amplitude, $T_{s}$ means the pulse duration, $f_{c}$ denotes the carrier frequency, and θ denotes the carrier phase. 

The mathematical expression of PSK signal can be expressed as:(14)$$x(t)=A\sum_{n}a_{n}g(t-nT_{s})\cos(2\pi f_{c}t+\varphi_{n}+\theta ),$$
where $\varphi_{n}$ denotes the modulation phase of the nth symbol, $\varphi_{n}\in \{\frac{2\pi m}{M},m=0,1,\cdots ,M-1\}$, *M* denotes the modulation order of the signal. 

The mathematical expression of QAM signal can be expressed as:(15)$$x(t)=\sum_{n}A_{k}g(t-nT_{s})\cos(2\pi f_{c}t+\varphi_{n}+\theta ),$$
where $A_{k}=\sqrt{A_{ki}^{2}+A_{kq}^{2}}$ denotes the carrier amplitude. 

The mathematical expression of LFM signal can be expressed as:(16)$$x(t)=A\mathrm{rect}\left(\frac{t}{T}\right)e^{j\left(2\pi f_{0}t+K\pi t^{2}\right)},$$
where *T* denotes the pulse width, *f*_0_ denotes the starting frequency, *K* denotes the frequency modulation slope, and $A\mathrm{rect}\left(\frac{t}{T}\right)$ is the rectangular function, which can be expressed as:(17)$$A\mathrm{rect}\left(\frac{t}{T}\right)=u\left(t+T\right)-u\left(t\right),$$
where u(t) is the step function.

The reason that 8 kinds of modulation are generated is to accommodate the detection of various radio signals for the detector. The candidate set consists of 4ASK, 8ASK, BPSK, QPSK, LFM, QAM16, QAM32, and QAM64. The training, validation, and testing sets accounted for 80%, 10%, and 10% of the dataset, respectively. We generate two categories of training data: one is the AWGN data, and the other is radio signals with AWGN noise.

To make the detector adapt to the environment with large scale of dynamic SNR, the radio signals are generated from −20 dB to 20 dB with an interval of 1 dB. The sample number of each modulation type is 4100, and the sample number of each SNR is 800, the sample number of modulated signal of each SNR is 100. Hence, the number of the dataset, training sample, validation sample, and testing sample is 32,800, 26,240, 3280, and 3280, respectively. The parameters of dataset in this work are shown in Table 2.

### 4.2. Experiments and Discussion

The effectiveness of the proposed algorithm is demonstrated through simulation experiments. In the simulation experiments, we consider that the channel gains obtained by each SU are independently and identically distributed. The “Average SNR” is the average SNR of the signal received by all SUs from the PU. The simulations are operated in the environment of the python 3.7 on GeForce RTX 2060.

In this section, we first explore the relationship between the number of model parameters (or floating point operations, FLOPs, in million) and accuracy for MCF. The details are shown in the Table 3.

Four schemes were designed according to the task characteristics of cooperative spectrum sensing, and cross-validation experiments were conducted to explore the relationship between the number of model parameters and accuracy. The more model parameters, the higher the accuracy. However, the difference between schemes 3 and 4 is rather insignificant. This indicates that the setting of model parameters for scheme 3 is more effective since scheme 3 has fewer parameters than scheme 4.

To explore the performance of MCN, we first trained the model with different number of nodes. Due to the limitation of the computing power of the equipment, two cooperative schemes with 2 and 4 nodes respectively are considered. We compare the detection performance of MCN with that of single-node spectrum sensing method in [33].

The detection performance of CSS schemes is illustrated in Figure 5. The performance of the cooperative sensing scheme based on MCN is superior to single-node sensing method. Furthermore, MCN (scheme 1) with 4 nodes achieve better detection performance than MCN (scheme 1) with 2 nodes. The simulation results show that the detection performance of MCN is obvious, especially under extremely harsh low SNR. As the SNR increased, the detection performance of MCN for CSS is more and more superior. MCF (scheme 1, node = 2) achieves the same performance as MCF (scheme 3, node = 4). Since MCF (scheme 1, node = 2) has more model parameters than MCF (scheme 3, node = 4), it’s more computationally expensive. Due to the MCF (scheme 3, node = 4) uses four channels representing four nodes, the multiscale information of signals is extracted and fused. Hence, it is proved that the MCF is effective without increasing the model parameters.

After the detection performance of MCN is verified by simulation, we conducted comparative experiments with other methods. For illustration purpose, the logical-and rule of Hard Decision is applied in the traditional cooperative sensing scheme, since it is currently the most widely used. According to GRU is more suitable for cooperative sensing tasks than LSTM, the verification experiment is proposed. We conducted experiments to compare the performance of MCF with other competitive methods, including CM-CNN [34], CL method [35], and CNN-LSTM, 1D-CNN [33].

Figure 6 illustrates the performance of various spectrum sensing schemes. The detection performance of MCN is superior to other spectrum sensing methods. When the number of nodes in the two cooperative systems is 4, the performance of MCN is better than HD. For simulations, GRU performs better than LSTM in cooperative sensing. When the SNR is lower than −15, thereby demonstrating the superiority of MCN.

Table 4 illustrates the parameters and FLOPs of involved models. The number of model parameters for MCN (scheme 1, node = 4) is more than other spectrum sensing methods. In general, the proposed model increases the model complexity, but also improves the accuracy. However, MCF (scheme 3, node = 4) gains superior results without increasing network parameters. CL Method connected CNN and LSTM in series and achieved effective results than CM-CNN.

To better compare the detection performance of the proposed CSS scheme with other schemes, we conducted experiments to analyze the receiver operating characteristic (ROC) curve. The ROC curve of different sensing schemes is summarized in Figure 7.

The purpose of analyzing ROC curve is to explore the relationship between $P_{d}$ and $P_{f}$. As the $P_{f}$ increased, the $P_{d}$ will rise significantly. When the $P_{f}$ is same, the $P_{d}$ of MCN (scheme 1) with 4 nodes is the largest.

In simulation, the generalization ability is an essential indicator of the model. We generated 8 kinds of modulated signals to test the well trained with 4 nodes. In particular, we conducted 8 experiments on the known 8 modulation mode signals, respectively. The performance of different modulation types is depicted in Figure 8.

Comparing the detection performance in [33], the generalization ability of MCN with 4 nodes is excellent. The performance of MCN is feasible even in low SNR. Even under the harsh conditions of −20 dB, the of $P_{d}$ BPSK can reach 93.9%. The performance difference between QAM16, QAM32, and QAM32 is rather insignificant. 

## 5. Conclusions

In this paper, we propose a Multifeatures Combination Network for cooperative spectrum sensing problem. The Multifeatures Combination Network extracts features through a parallel structure, which exploits the complementarity of modelling capabilities of CNN and GRU. Moreover, the parallel structure extracts spatial features and temporal features simultaneously and concatenates them in Combination-net. The Multifeatures Combination Network improve the reliability of sensing by fusing the perception information of multiple nodes. Furthermore, comparing conventional cooperative spectrum sensing schemes delivered the final result, based on the binary decision result, the Multifeatures Combination Network obtains final decision by utilizing the confidence information of each node. The simulation results show that the detection performance of Multifeatures Combination Network with 4 nodes is superior to other methods, which demonstrates its effectiveness especially under harsh sensing conditions. Eventually, results demonstrate the generalization ability of the proposed model for multiple modulation types. The results show that our proposed Multifeatures Combination Network achieved competitive performance. Although our proposed method can achieve efficient performance, the computational complexity is not considered as the focus of discussion. The computational complexity plays a critical role in the practicability of spectrum sensing methods. In the future, we will focus on reducing computational complexity while maintaining the validity of the method. Another future work is to apply the Multifeatures Combination Network to a more complex and larger dataset of real signals. Once it is further shown that it can achieve effective performance in a real environment, the superiority of our method will be demonstrated.

## Figures and Tables

**Figure 1 entropy-24-00129-f001:**
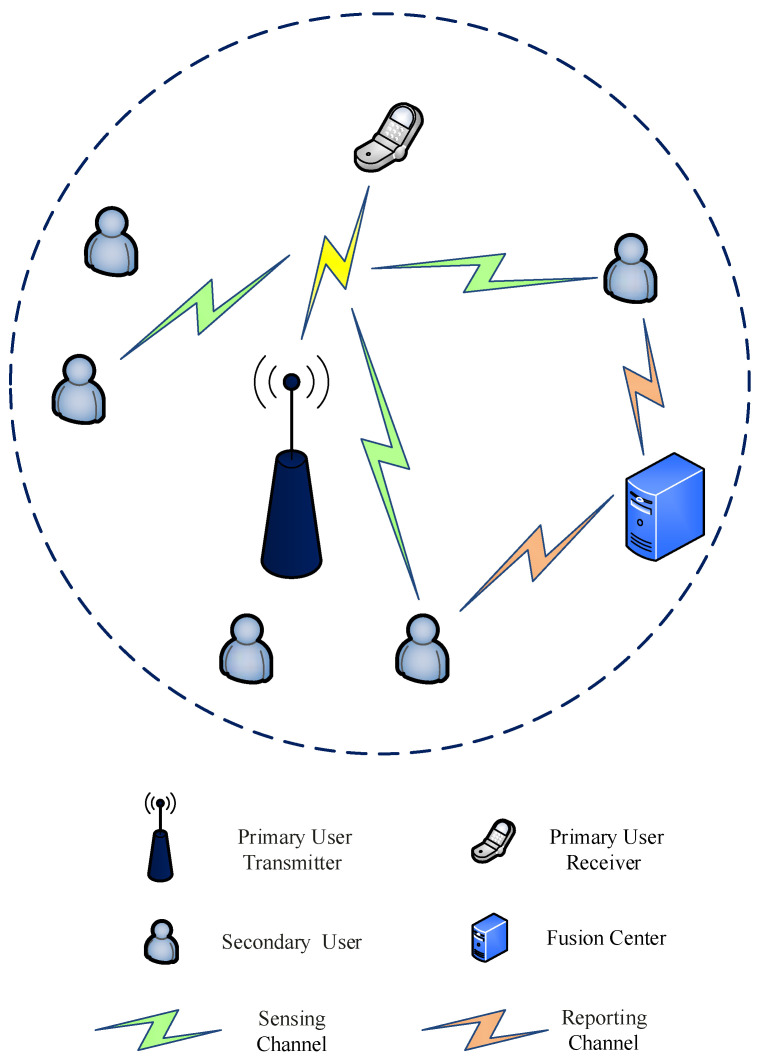
System model for cooperative spectrum sensing.

**Figure 2 entropy-24-00129-f002:**
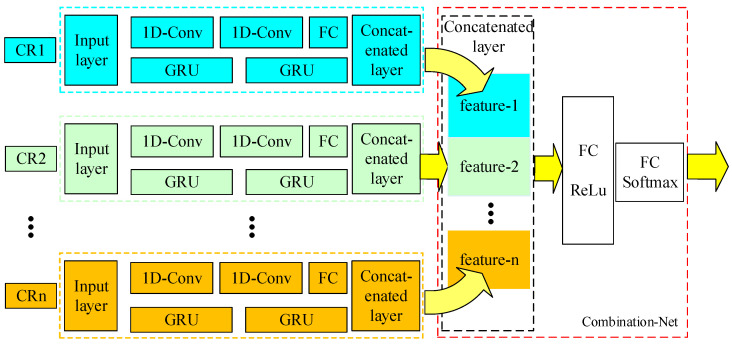
Architecture diagram of Multifeatures Combination Network (MCN) for cooperative spectrum sensing.

**Figure 3 entropy-24-00129-f003:**
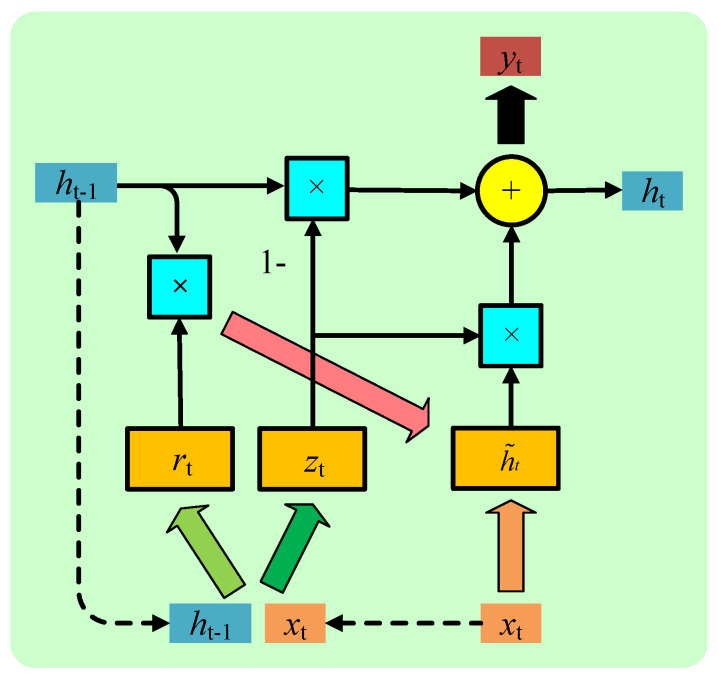
Architecture diagram of gate recurrent unit (GRU) for cooperative spectrum sensing.

**Figure 4 entropy-24-00129-f004:**
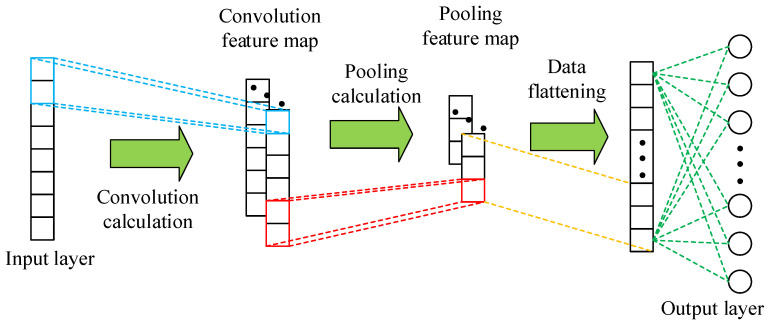
Schematic diagram of one-dimensional convolution neural network (1D-CNN) for cooperative spectrum sensing.

**Figure 5 entropy-24-00129-f005:**
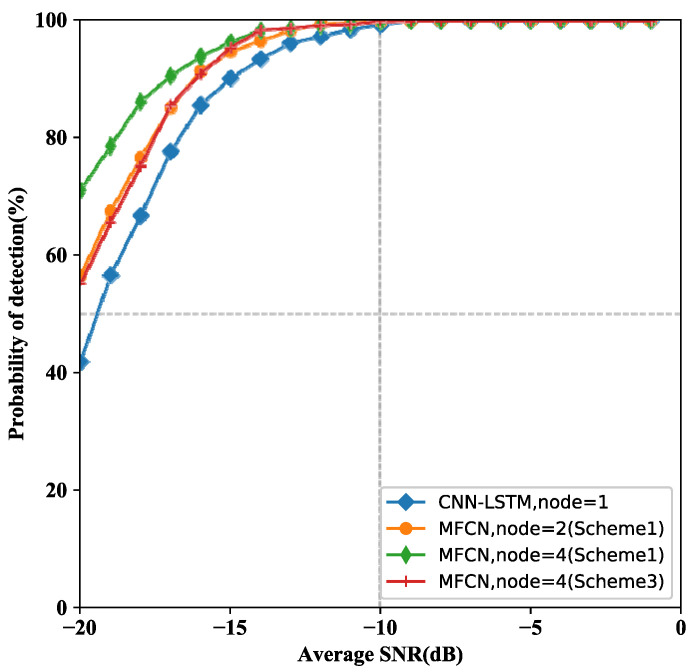
Detection performance for models with different number of nodes.

**Figure 6 entropy-24-00129-f006:**
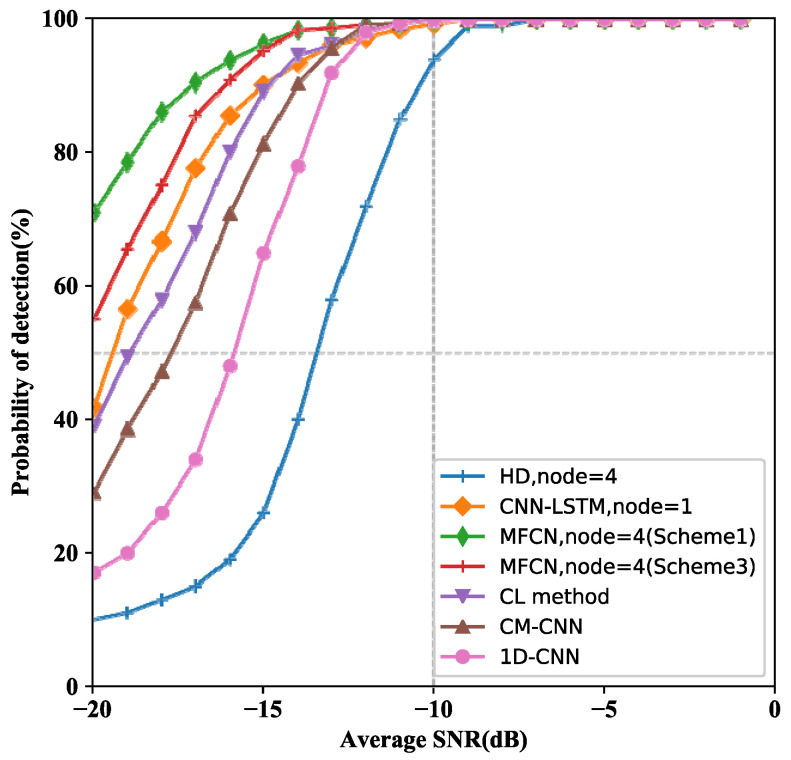
Detection performance for various spectrum sensing methods.

**Figure 7 entropy-24-00129-f007:**
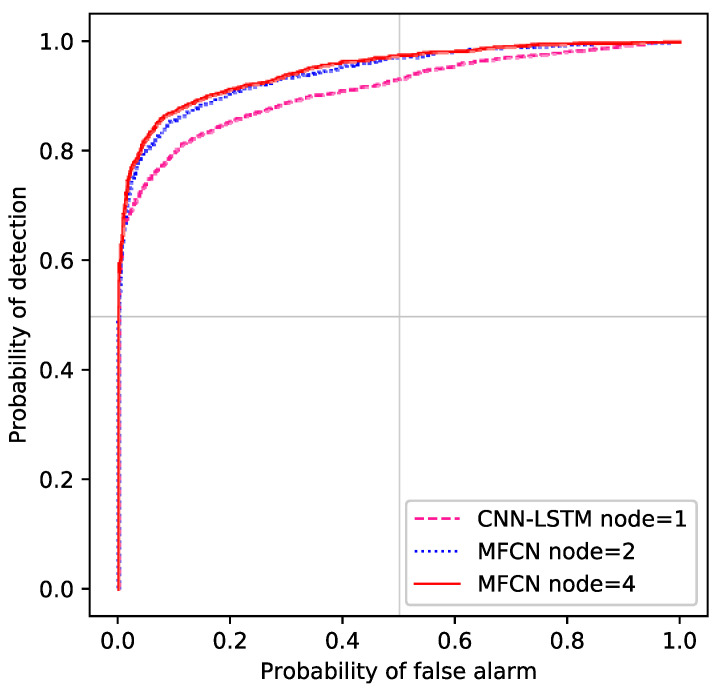
Receiver operating characteristic (ROC) curves for different number of nodes at SNR = −18 dB.

**Figure 8 entropy-24-00129-f008:**
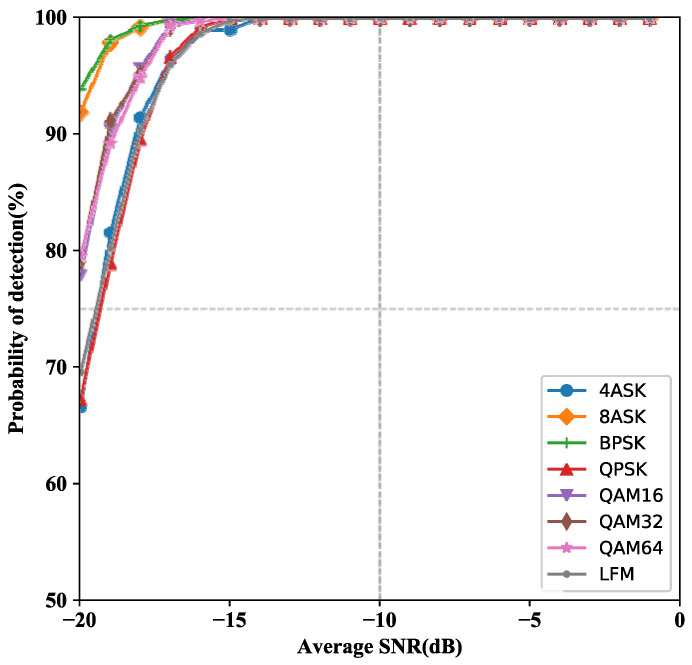
Performance of different modulation types for MCN (scheme 1) with 4 nodes.

**Table 1 entropy-24-00129-t001:** Hyperparameters of MCN.

Hyperparameters	Value
Filters per FC layer of Combination-net	8 & 2
Batch size	128
Epoch	50
Dropout ratio	0.33
Optimizer	Adam
Initial learning rate	0.0005

**Table 2 entropy-24-00129-t002:** Parameters of dataset.

Parameter	Value
SNR range	−20~20 dB
Modulation categories	4ASK, 8ASK, BPSK, QPSK, QAM16, QAM32, QAM64, LFM
Dataset samples	32,800
Samples per type	4100
Samples per SNR	800
Samples per type of per SNR	100
Training samples	26,240
Validation samples	3280
Testing samples	3280

**Table 3 entropy-24-00129-t003:** Different schemes of MCF.

		Value			
		Scheme 1	Scheme 2	Scheme 3	Scheme 4
Convolution kernels per 1D-Conv layer	CR1	128 & 32	32 & 8	16 & 8	16 & 16
	CR2	64 & 32	16 & 8	8 & 8	8 & 16
	CR3	32 & 32	8 & 8	4 & 8	4 & 16
	CR4	16 & 32	4 & 8	2 & 8	2 & 16
Filters of FC layer	CR1	16	4	4	4
	CR2	16	4	4	4
	CR3	16	4	4	4
	CR4	16	4	4	4
Hidden nodes per GRU layer	CR1	128 & 32	32 & 8	16 & 8	16 & 16
	CR2	64 & 32	16 & 8	8 & 8	8 & 16
	CR3	32 & 32	8 & 8	4 & 8	4 & 16
	CR4	16 & 32	4 & 8	2 & 8	2 & 16
Number of perparameters	Node = 4	0.40 M	0.13 M	0.12 M	0.13 M
	Node = 2	0.26 M	0.07 M	0.06 M	0.07 M
FLOPs	Node = 4	125.38 M	9.64 M	5.30 M	9.77 M
	Node = 2	98.36 M	7.43 M	3.90 M	7.17 M
Loss	Node = 4	0.143	0.164	0.167	0.168
Test accuracy	Node = 4	95.6	94.7	94.4	94.0

**Table 4 entropy-24-00129-t004:** Comparisons of parameters and FLOPs of involved models.

Model	Parameters	FLOPs
CM-CNN [34]	1.19 M	7.11 M
CL Method [35]	0.20 M	0.45 M
CNN-LSTM [33]	0.22 M	1.05 M
MCF, Node = 4 (scheme 1)	0.40 M	125.38 M
MCF, Node = 4 (scheme 3)	0.12 M	5.30 M

## Data Availability

Some or all data, models, or code that support the findings of this study are available from the corresponding author upon reasonable request.

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
