# Peer review of "Cooperative Spectrum Sensing Based on Multi-Features Combination Network in Cognitive Radio Network"

_entropy, 2022, doi:10.3390/e24010129_

Round 1
Reviewer 1 Report
The paper presents a hybrid convolutional neural network (CNN)/gate recurrent unit (GRU) for cooperative spectrum sensing. The paper has a few serious issues:
1) The authors try to "sell" the idea of CNN/GRU of their own. However, there have been numerious papers dealing with CNN/GRU also for spectrum availability prediction, not mentioned in the submitted manuscript such as the general work with DOI: 10.1109/ICKII.2018.8569155 by Lee and many more. and the particular work "Spectrum Availability Prediction for Cognitive
Radio Communications: A DCG Approach" by Lixing Yu et al, DOI: 10.1109/TCCN.2020.2973572. It looks like that the latter does pretty much the same, but published 2 years ago. The authors have to clarify the novelty with respect to existing approaches.
2) For the GRU unit, equations 6 and 8 are generally speaking wrong. This is because the same matrix cannot be applied to the input vector x and the output vector h. What kind of (strong) assumnptions have been done to accomplish this simplification and what is the impact with respect to the general solution?
3) With the current explanation in the manuscript, Fig.5. and 6 are wrong. There is no way to receive correct packages at -20dB SNR (per bit, byte, packet??)
How and where exactly in the chain is the "average SNR" defined? What are the assumptions on the wireless channel? What kind of redundancy has been inserted in the communication link. (coding, modulation,...) Without redundancy communication stops around 0 dB SNR per bit at the output of the slicer according to Shannon. Also unclear is over what exactlly has been averaged in the SNR (PU and SU nodes, channel, ...)
Specific comments.
English language is quite poor.
Abstract: ... gate recurrent unit networks.... no, GRU are a gating mechanism in recurrent neural networks.
Section 3.1. We proposed the ... has been proved to be feasibile. Active and passive cannot mixed together. Moreover, proved -> proven.
Section 3.1: Through extensive cross-validation,
the concatenated layer with 32 neurons and two FC layer with 8 and 2 neurons, respectively. This sentence does not make sense. Some verb is missing
Section 3.2: .. has been appeared powerless in solving the input sequence.. What does this sentence mean. Reformulate!
Section 3.2: ... which by inbtroducing the gate structures. Which does what? The sentence ends unfinished.
Section 4.2. "...with different number of node" node-> nodes.
Author Response
We would like to thank you for these precious comments concerning my manuscript. These comments are all valuable and very helpful for revising and improving my paper, as well as the important guiding significance to my researches. We have studied comments carefully and have made corrections which we hope meet with approval. Please see the attachment. Thank you.

Reviewer 2 Report
In the manuscript, the authors aim to deal with the cooperative spectrum sensing in cognitive radio network using CNN-GRU. Eventhough the idea in this manuscript is interesting, I don't think it is quite novel and sufficiently evaluated. My main concerns are as follows:
- Title: the deep learning term in the title is too vague, it should be mentioned precisely as the CNN-GRU or others.
- Abstract: what is the so called Feature Combination Network here? what is the problem in cognitive radio network you exactly want to solve?
- Page 4: what is "Kth SU"?
- In this manuscript, their proposed Feature Combination Network is abbreviated as FCN, which is misleading as the FCN has been well known for the Fully Connected Network. Could you find another appropriate abbreviation?
- The author only used merely one single generated dataset. It is not sufficiently convincing. Could you use some real datasets to test your proposed model so it is evaluated scientifically?
- I could not find the scalability evaluation and discussion of the proposed CNN-GRU model? Is it computationally expensive to train?
Author Response

(The authors gave the same response as above.)

Reviewer 3 Report
This paper is dedicated to the presentation of a cooperative spectrum sensing algorithm based on convolutional neural networks and gate recurrent unit. The subject can be a matter of interest for the community. The following issues need to be considered in the revised version of the manuscript.
1- The content of the paper scientifically makes sense. However, the authors should elaborate some of the technical sections with more details. For instance, in Subsection 2.2, references should be provided for the model of spectrum sensing.
2- Authors should provide reasonable justification for considering the hyper parameters in Table 1.
3- Fig. 5 to 8 are better to be provided in vector format, similar to the rest of the figures.
4- The conclusion section lacks the potential future works for this research.
Author Response

(The authors gave the same response as above.)

Round 2
Reviewer 1 Report
The major problem of the new version is that Fig.5, 6 and 8 are kept the same and hence, are kept wrong. For example, the authors claim that BPSK (green curve) in Fig.8 has 0 error probability at SNR (averaged over all SUs) equal zero. According to the reply of the authors, there has not been implemented any sort of redundancy in the transmission scheme. In AWGN without interference, the bit error probability of BPSK would be ca. 0.2, corresponding to a probability of correct detection equal 0.8 only -- in contrast to what is stated in the manuscript. At -20dB, detection is impossible in contrast to what is stated. When the signal of 4 nodes is received, performance can only become worse. Analogous can be said for the other modulation schemes, appearing also in Fig. 5 and 6. Hence, the results are definitely wrong.
The new version of the manuscripts states that the novelty is the "[31,32] ... parallel CNN-GRU structure while previous versions used a serial structure" (line 149 ff). This statement is without proof. How much is the gain of the proposed solution with respect to the "serial approach" in [32]? Is there is any?
It is recommended to redo the simulations with the proper SNR definition, add the performance of the competitive scheme [32], point out the gain of the proposed solution with respect to [32] and resubmit the paper.
Author Response

(The authors gave the same response as above.)

Reviewer 2 Report
I appreciate the authors' effort in responding to my concerns.
However, my previous two biggest concerns have not been sufficiently addressed yet.
1. The performance evaluation on merely one single generated dataset is too far from scientifically sufficient. You could not hide behind a short explanatory paragraph. There are at least two ways to prove that our proposed method is worth considering. First, theoretically/mathematically proven, which have not been done in the manuscript and I am not sure whether this manuscript will go for this kind of research. Second, empirically evaluated in a systematic approach, which I expect from this manuscript. If the real datasets are impossible to get, you may try a systematic evaluation using multiple different generated datasets with different characteristics. To this end, we may assume that the reported results are not merely a bit of luck or a random coincidence.
2. The scalability evaluation and discussion is a must. It can be considered a lazy experiment to have no scalability evaluation and discussion. It will be useful to have a systematic scalability evaluation of the parameter (or hyperparameter) sensitivity of the proposed method.
Author Response

(The authors gave the same response as above.)

Reviewer 3 Report
No comments
Author Response
We would like to thank you for these precious comments concerning my manuscript. These comments are all valuable and very helpful for revising and improving my paper, as well as the important guiding significance to my researches. Thank you.
Round 3
Reviewer 1 Report
The authors have tackled all open issues.
Reviewer 2 Report
OK